# Corporate Social Responsibility and Corporate Performance: A Hybrid Text Mining Algorithm

**Mushang Lee [1] and Yu-Lan Huang [2],***

1  Department of Accounting, Chinese Culture University, Taipei 11114, Taiwan; leemushang@gmail.com
2  Graduate Institute of International Business Administration, Chinese Culture University, Taipei 11114, Taiwan
*  Correspondence: nancyhuangyulan@gmail.com; Tel.: +886-2-2861-0511 (ext. 35512)

**Abstract:** Until now, the works regarding the relationships between corporate operating performance and corporate social responsibility (CSR) could not reach a conclusive result (positive, natural, and negative). This circumstance can be attributed to two main reasons: (1) inadequate performance measurement and (2) ignoring the multi-dimensional nature of CSR. To combat this, we provided a hybrid decision framework that consisted of two main procedures: (1) performance measurement via linear programming algorithm and (2) CSR's multi-dimensional nature extraction via text mining. By joint utilization of a linear programming algorithm and text mining, we could gain more insights from the outcome. The proposed decision framework, tested by real cases, is a promising alternative method for performance prediction. Managers can take this model as a roadmap and allocate resources to suitable places, as well as reach the goal of sustainable development.

**Keywords:** corporate social responsibility; decision-making; performance; text mining

## 1. Introduction

Market crises and accounting scandals like Black October 1987, the 1997 Asia financial turmoil, and the subprime mortgage debacle in the U.S. have received tremendous attention from academia and practitioners, as is evident from current financial literature [1]. These extraordinary and unusual events have had fatal side effects or caused damage not only to the respective financial markets, but also to societies as a whole. Due to such practical significance, a considerable amount of research has looked into corporate financial crisis prediction [2,3].

Preliminary works on financial pre-warning models were frequently based on statistical approaches. Although statistical approaches have satisfactory prediction capabilities, they also confront numerous challenges, such as linear separability, multivariate normality, and independent predictor variables, which do not hold up in real-life applications. With the great improvement in information technology, numerous artificial intelligence (AI) techniques that do not obey strict statistical assumptions have been introduced to solve the aforementioned problems [4–6]. In contrast to well-examined issues such as credit rating forecasting and financial crisis prediction, research on corporate operating performance forecasting is rather scant. It is, however, widely acknowledged that the main cause of financial crises on the corporate end is poor management, and operating performance is a suitable and reliable proxy for representing the outcome of corporate management [7–9]. Kamei [10] also stated that 99% of financial crises are due to bad operating performance—that is, the decline in operating performance can be viewed as a prior stage before a financial crisis bursts forth.

Return on assets (ROA) and return on equities (ROE) are the most commonly executed performance measures for a firm [11,12]. As these measuring criteria are categorized into one input variable and one output variable, they cannot suitably represent the whole facet of a corporation's operations in

a turbulent economic atmosphere. Data envelopment analysis (DEA) can deal with this obstacle, as it can handle multiple input and multiple output variables without an a priori assumption of profit maximization or cost minimization [13]. It can also be extended into different modifications and provide a synthesized outcome, thus offering adequate flexibility for practical applications [14–16]. How to precisely measure a corporation's operating performance and further construct a reliable forecasting model are notable topics for decision makers [17].

Stein [18] stated that lower capital constraints may be one possible avenue to enhance corporate operating performance, because these constraints have a direct impact on a corporation's ability to undertake major investment decisions and capital structure selections. Benabou and Tirole [19] revealed that a firm with superior corporate social responsibility (CSR) performance can end up with lower capital constraints. The reasons can be summarized as follows. First, better CSR performance is connected to superior stakeholder engagement, which can eliminate short-term risky and opportunistic investment strategies and, as a consequence, decrease whole contracting costs [20]. Second, a firm with better CSR performance, it can spread positive messages to all market participants to signal its long-term focus [19,21]. Finally, better CSR performance is also linked to data accountability and transparency, thus alleviating the obstacle of information asymmetry between managers and investors and leading to lower capital constraints [22–26].

Friedman [27] conversely viewed CSR as an agency problem and indicated that CSR has a negative effect on operating performance due to corporate costs from CSR. Based on the agency cost theory, Brown et al. [28] stated that top executives may benefit themselves utilizing their corporations' inherent resources through philanthropy while shareholders incur a loss by such spending on charity. Barnea and Rubin [29] also indicated that top executives overinvest in CSR to build up their personal reputation when agency costs are incurred. Despite the large amount of concentration, a basic question remains unanswered: Does CSR lead to value creation (i.e., enhance corporate operating performance)? If so, in what ways? The extant studies so far have failed to give a conclusive answer [30].

Cavaco and Crifo [31] stated that one possible reason for this absence of consensus lies in the fact that CSR policy is multi-dimensional and consists of social, environmental, and business behavior facets. Merely implementing a singular item as a proxy for generic CSR could lead to a confused result about the relationship between CSR and operating performance [32]. Obviously, there is an urgent requirement to break down CSR among dissimilar dimensions so as to realize its possible differential influence on corporate operating performance [33,34]. To deal with the aforementioned challenge, the topic extraction algorithm, called latent Dirichlet allocation (LDA), was formed [35]. After executing the LDA algorithm, one can extract the informative and interpretable topics/dimensions from CSR reports and then examine the effectiveness of each topic/dimension on operating performance.

The valuable topics/dimensions extracted by LDA and performance rank determined by DEA can then be injected into an extreme learning machine (ELM) (one of the neural network-based (NN-based) techniques) to construct the forecasting mechanism. In contract to a traditional NN-based algorithm, an ELM not only reaches superior generalization ability with an efficient calculation speed (i.e., its input weights and hidden biases are arbitrarily decided, and the output weights are computed by performing a Moore–Penrose (MP) generalized inverse), but also avoids many troubles in manually tuning inherent parameters, such as stopping criteria, learning rate, learning epochs, and local minima [36–38].

To our knowledge, extant works have not yet established a forecasting model for operating performance that simultaneously integrates numerical information from financial reports and textual information from CSR reports. This study thus further decomposes CSR into numerous dimensions to realize the effect of each dimension on performance forecasting. By doing so, top executives can realize which dimension has a great effect on operating performance and then allocate valuable resources to suitable places to reach the goal of sustainable development.

We believe that the results herein contribute to the corporate finance and CSR literature in several ways. First, this study proposed a novel mechanism to forecast operating performance, which is widely acknowledged as the prior stage to any financial crisis. Second, rather than simply investigating the

influence of generic CSR on corporate operating performance, we broke down CSR into numerous dimensions by a topic extraction technique and then examine the effectiveness of each dimension/topic on corporate operating performance. As the CSR dimension was determined by a data-driven technique (i.e., that is, a topic extraction technique tries to let words speak), the result is less reliant on human involvement and more objective. The CSR measures in this study were totally different from previous studies that focused on developing the measurement of a theoretical construct. The study provides a different viewpoint to discuss the impacts of CSR on performance. Finally, employing a Taiwanese database was appealing, because most practical works on CSR and corporate operating performance assessment have focused on developed economies (such as corporations in the U.S. and EU) with less agency cost between manager and shareholders, and, thus, a positive effect of CSR on performance is generally supported. Firms in developing economies (such as Taiwan) and developed economies have distinctive differences in organizational structures and behaviors [39,40]. Lin [41] also stated that the ownership structure in Taiwan is less dispersed than it is in the United States and United Kingdom—that is, the agency problem in developing economies is much more severe than in developed economies. Thus, this study used results in Taiwan to investigate the relationship between CSR and performance.

This paper is organized as follows. It starts with research methodologies adopted for this study, followed by research design, data illustration, and practical results. This paper closes with conclusions and implications for future research.

## 2. Methodologies

### 2.1. Data Envelopment Analysis

Introduced by Charnes et al. [42], DEA is a widely used economic analysis quantitative approach to assess the relative efficiency of a homogeneous set of decision-making units (DMUs) with multiple input and output variables. A performance comparison of the DMUs is conducted through an assessment of the differences in their input and output variables [43]. An input index can utilize consumption, such as economic resources, and labor hours; an output index represents output outcomes, such as sales revenues, and productivity. By integrating the input and output indices, we can make a further comparison of each DMU's economic benefit.

The synthesized index is set up by assigning a weight to these individual indices and then aggregating the weights. The weights of each input variable and output variable are calculated by optimizing the relative efficiency. This is the reason why DEA can handle a problem with uncertain weights of input and output indices and one that contains a large number of objectives in each DMU's performance assessment. The BCC (Banker–Charnes–Cooper) DEA model was executed in this study as the corporate operating condition often achieves variable return to scale (VRS), due to imperfect competition, government regulation, and financial constraints. In addition, an output orientation was conducted because once a corporate financial investment decision has been made, it is very complicated to disinvest to save the costs by amending input variables, thereby invalidating the output orientation.

The output-oriented BCC DEA model evaluates the relative efficiency of $n$ corporations ($DMU_k$, $k = 1, \ldots, n$). Every $DMU_k$ uses $p$ inputs ($I = 1, \ldots, p$) and generates $q$ outputs ($r = 1, \ldots, q$). The relative performance score of $DMU_k$ can be described as follows:

$$
\begin{aligned}
Max\ h_k &= \sum_{r=1}^{q} u_r y_{rk} - u_b \\
s.t.\ &\sum_{i=1}^{p} G_i x_{ik} = 1 \\
&\sum_{r=1}^{q} u_r y_{rj} - \sum_{i=1}^{m} G_i x_{ij} - u_b = 0, \\
&u_r, G_i \geq \delta,\ i = 1, \ldots, p \\
&r = 1, \ldots, s,\ and\ u_b\ free\ in\ sign,
\end{aligned}
\tag{1}
$$

where $h_k$ is the performance score of corporation $k$; $y_{rj}$ denotes the $r$-th outputs of the $j$-th DMU; $x_{ij}$ represents the $i$-th inputs of the $j$-th DMU; $u_r$ describes a weight of $r$-th output of corporation $k$ and $v_i$ denotes a weight of $i$-th input of corporation $k$. In addition, $\delta$ expresses the extremely small positive number to make all $u_r$, $v_i$ positive; $u_b$ denotes intercept. Based on the above-mentioned model, the optimal input/output multiplier can be decided [44].

### 2.2. Topic Modeling: Latent Dirichlet Allocation (LDA)

Latent Dirichlet allocation (LDA) is an unsupervised probabilistic generative model for describing the latent topics of documents [35]. It models each document as a distribution over topics and each topic as a distribution over words. These distributions are followed by the Dirichlet distribution. Let $U$ and $B_u$ represent the set of users and the bag of words determined by a user $u \in U$, respectively. Let $W$ be the set of specific words showing up in a bag of words $B_u$ at least once for a user $u \in U$. Here, $H$ expresses the set of latent topics, and the volume of topics is given as a parameter. In LDA's generative procedure, each user $u$ has his/her own preference over the topics expressed by a probabilistic distribution that is a multinomial distribution over $H$, denoted by $\vec{\theta}_u$, and each topic $h$ also has to follow a multinomial distribution over $G$, expressed by $\vec{\theta}_h$.

Figure 1 expresses the generative procedure of LDA, represented in a graphical style. In Figure 1, circles are implemented to represent the random variables or model parameters in the LDA model. Shaded circles denote the variables or parameters that are observable in the dataset, and the white circles express the unobservable ones that have to be estimated by inferencing mechanisms. The arrows depict the dependencies between variables. We express LDA's generative procedures below.

For each topic $h \in H$.

Draw a multinomial distribution $\phi_h \sim Dirichlet(\vec{\beta})$

For each user $u \in U$.

Draw a multinomial distribution $\theta_u \sim Dirichlet(\vec{\alpha})$

For each word $v \in B_u$

(a) Draw a topic $z \sim Multinomial(\vec{\theta}_u)$

(b) Draw a word $v \sim Multinomial(\vec{\phi}_h)$

In the LDA model, the multinomial distributions $\vec{\theta}_u$ and $\vec{\phi}_h$ are sampled from the Dirichlet distribution, there is one kind of conjugate prior distribution, and the inherent parameter of $\vec{\theta}_u$ and $\vec{\phi}_h$ are $\vec{\alpha}$ and $\vec{\beta}$, respectively. Each word $v$ in $B_u$ is presumed to be chosen by first drawing a topic $h$, following the topic preference distribution $\vec{\theta}_u$, and subsequently selecting a word $v$ from the corresponding distribution $\vec{\phi}_h$ of the selected topic $h$. According to the definition of the LDA model, the following equation estimates the probability that a word $v$ is constructed by a user $u$.

$$\int Dirichlet(\theta_u; \alpha) \left( \sum_{h=1}^{H} \theta_{uh} \phi_{hv} \right) b\theta_u \tag{2}$$

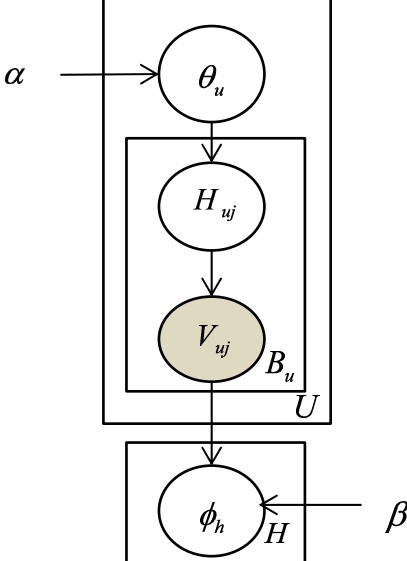

**Figure 1.** The graphical expression of latent Dirichlet allocation (LDA) model [45].

### 2.3. The Proposed Model: LDA_ELM_SA

This study introduced an emerging hybrid mechanism, the LDA_ELM_SA model, for operating performance forecasting, which consisted of four main procedures: (1) establishing a CSR-related corpus by a text mining (TM) technique, (2) extracting CSR's multi-dimensional characteristics by LDA, (3) determining the important features by fuzzy rough set theory (FRST), and (4) constructing a forecasting model by ELM with the self-adaptive mechanism.

- Procedure 1:

In order to handle a tremendous amount of CSR textual information and further condense it into manageable factors, a CSR-related corpus should first be decided. To reliably and soundly establish the CSR-related corpus, we had to gather CSR reports from two different categories: (1) corporations that have received a CSR award (that is, those with good CSR performance), and (2) corporations that have not received a CSR award, or corporations that have violated CSR rules, been fined for environmental pollution, legal problems related to abusing employees, or some social issues (that is, those with bad CSR performance). The CSR award is announced by CommonWealth Magazine, which is one of Taiwan's most prestigious publishing groups, and is based on four core subjects: corporate governance, enterprise commitment, community involvement and development, and environmental protection.

Li [46] indicated that financial reports contain a large number of optimistic terms that could represent a corporation's chances of gaining profit in the following years. Based on this concept, we collected CSR reports from two categories and then performed Chi-square to identify the representative special terms from two distinctive categories—that is, the terms in two different categories are quite different. How to quantify the importance of each special term is another interesting topic for users. The entropy technique, which has been commonly performed in computational linguistic analysis as a measure of the relative weights among all special terms, was conducted. As corporate CSR reports are guided by Global Reporting Initiatives (GRI), terms used in the reports are very specific and precise. Thus, the result from the entropy method is reliable.

- Procedure 2:

In order to deal with the nature of CSR's multi-dimensional characteristics, LDA was performed. LDA is a generative model that can be implemented to a corpus of documents made up of categorical data [35,47,48]. In this algorithm, each document is expressed as a random mixture of latent topics, which are multinomial distributions over the unique words in a corpus [49]. The generative procedure for a document includes determining a topic distribution that satisfies the corpus-level Dirichlet

distribution. Sequentially, for each word embedded into the document, a topic is decided and a word is picked up from the corresponding multinomial distribution. After executing the LDA algorithm, we could decompose CSR into some essential dimensions and further match the preserved dimensions with a pre-determined CSR corpus so as to condense a large amount of textual information into manageable elements. By doing that, we could examine which dimension had a great effect on performance.

- Procedure 3:

The collected information was usually full of irrelevant, redundant, and useless material, which in most situations would mislead the research findings, obtain biased results, and increase computational burdens. Feature subset selection is an inevitable prior stage before forecasting model construction. The purpose of feature selection is to remove redundant or useless features without significantly deteriorating the forecasting performance of the model constructed merely by the selected feature subsets being as close as possible to the original class distribution [50]. Rough set theory (RST) was proposed by Pawlak [51] as a tool for handling data with uncertainty, vagueness, and incompleteness and has been applied successfully to feature subset selection and rule induction in numerous research fields [52]. However, the classical RST is defined with equivalence relations that impede its capability at handling data with numerical or fuzzy values [53]. Discretization of the numerical data is one avenue to solve this challenge, but it may result in the problem of information loss. In order to cope with categorical and numerical data in datasets, fuzzy rough set theory (FRST) was introduced by Dübois and Prade [54] through a combination of RST and fuzzy set theory (FST) together. In this approach, a fuzzy similarity relation is executed to assess the similarity between samples. The fuzzy upper and lower approximations of a decision are then determined by utilizing the fuzzy similarity relation [55]. The fuzzy positive region is denoted as the union of the fuzzy lower approximations of decision equivalence classes. As the fuzziness is introduced to RST, more valuable information of real-valued data is easily preserved. Feature subset selection with FRST has gained considerable attention in recent years [56]. Thus, FRST was conducted in this study. For a more detailed illustration of FRST, please refer to Jensen and Shen [57].

- Procedure 4:

The selected features by FRST were finally executed to construct the corporate operating performance forecasting model. In traditional ELM, the number of hidden nodes is pre-decided, and the hidden node parameters are arbitrarily decided and remain unchanged during the training process, resulting in the existence of non-optimal nodes and a failure to reach the aim of cost function minimization [58]. In order to overcome the aforementioned problems, ELM with a self-adaptive (ELM_SA) mechanism was proposed. In ELM_SA, the hidden node learning parameters are optimized by a self-adaptive differential evolution approach, whose trial vector construction strategies and their associated control parameters are self-adaptive in a strategy pool by learning from their prior experiences in providing a promising solution, and the network output weights are computed via the Moore–Penrose (MP) generalized inverse. To ameliorate the impact of over-fitting, five-fold cross-validation was conducted. Figure 2 depicts the flowchart of the proposed LDA_ELM_SA model.

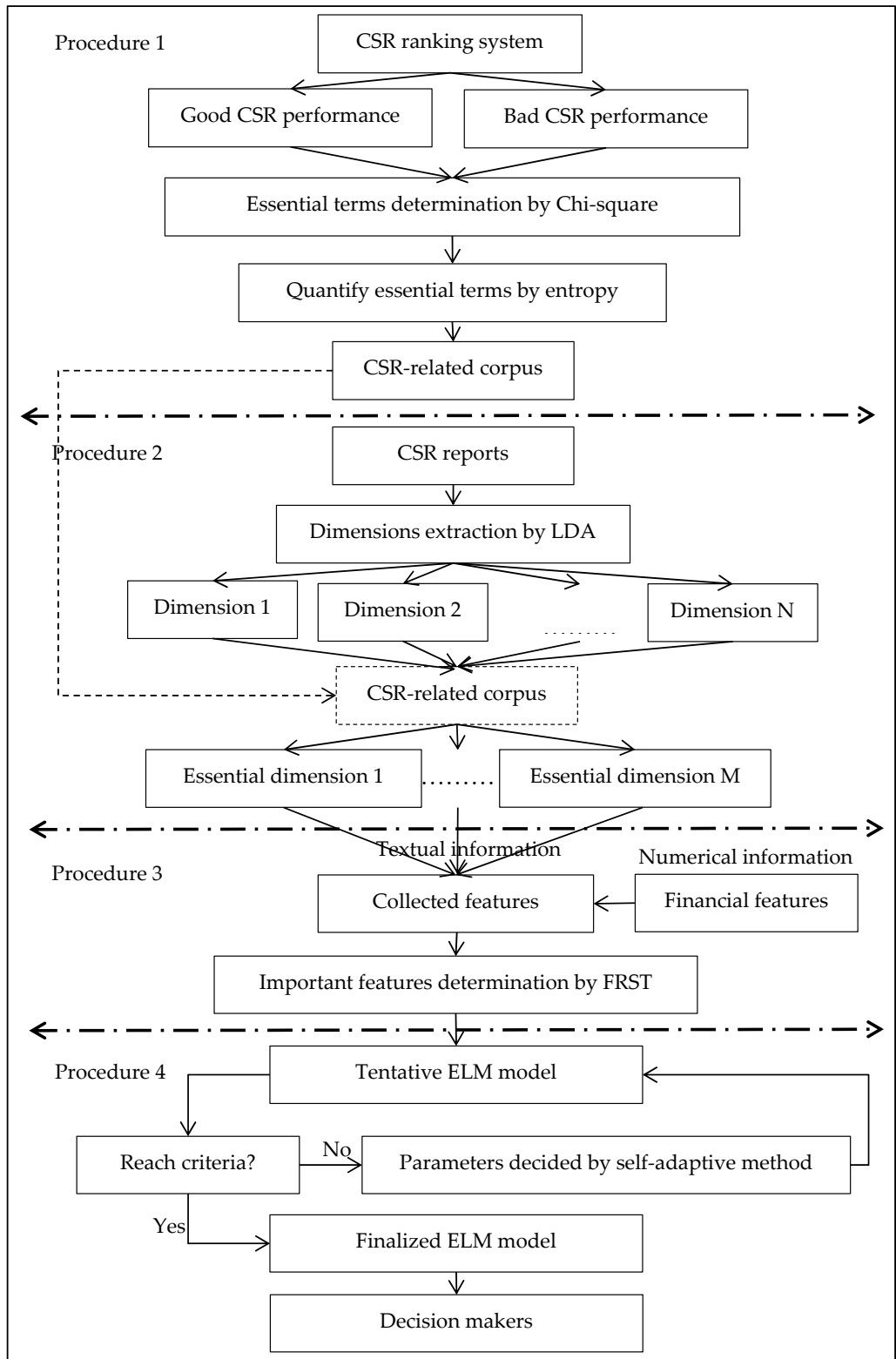

**Figure 2.** The flowchart of LDA_ELM_SA model. Abbreviations: CSR—corporate social responsibility, FRST—fuzzy rough set theory, ELM—extreme learning machine, SA—self-adaptive.

## 2.4. The Empirical Question

After going through the aforementioned procedures (see Figure 2), we introduced the model. The performance status (i.e., dependent indicator) was determined by DEA. The independent indicators were mainly collected from two parts: (1) quantitative predictors derived from financial statements, and (2) qualitative predictors collected from CSR reports. Our study examined whether the CSR textual reports contained supplementary information beyond what numerical indicators could reveal. Therefore, we proposed the following hypothesis.

**Hypothesis 1 (H1).** *The performance forecasting model with textual indicators has higher forecasting capability than the traditional numerical models.*

## 3. Research Results

### 3.1. The Sample

Taiwan has not only been admired as an economic miracle of the world, but also has tremendous impacts on the global electronics supply chain and its capital market plays an essential role for global stock market investors. Among all of the industries in Taiwan, the electronics industry has caught the most attention from researchers. It is because the public sectors or central government have announced many financing or investment incentives and allocated tremendous economic resources to this specific industry, turning it into a mainstay of the domestic stock market. Moreover, stock transactions in the electronics sector make up over 70% of Taiwan's stock market turnover. Thus, this specific industry was taken as our research target, with the data gathered from public websites and databases, such as the Taiwan Stock Exchange (TWSE), Taiwan Economic Journal (TEJ) databank, and Taipei Exchange (TE) from the period 2016–2018.

### 3.2. The Indicators

Dependent indicator—this study set up a model to forecast which operating performance indicators of corporations belong to the efficient sample and which belong to the inefficient sample. How to appropriately discriminate the efficient samples from inefficient samples is an interesting task. We used DEA to handle the aforementioned task, because it has proven remarkably popular for conducting performance measurement tasks in recent decades. In accordance with previous related works [8,15,16], we chose total assets (TA) and total liabilities (TL) as input variables and designated three profitability measures, return on assets (ROA), sales growth rate (SGR), and profit margin (PM), as output variables. To test the representativeness of the chosen variables, the Pearson correlation was performed.

Table 1 presents the results. We can see that all the chosen variables have a positive correlation—that is, no chosen variables should be omitted. Sequentially, we arranged corporate operating performances from efficient to inefficient guided by the DEA's synthesized value and further designated the top 20% as the efficient category and the bottom 20% as the inefficient category. Doing so helped us to transform the forecasting problem into a classical binary classification task.

**Table 1.** The result of Pearson correlation.

| Variables | TA | TL | ROA | SGR | PM |
|:---:|:---:|:---:|:---:|:---:|:---:|
| TA | 1 | – | – | – | – |
| TL | 0.905 | 1 | – | – | – |
| ROA | 0.477 | 0.328 | 1 | – | – |
| SGR | 0.503 | 0.438 | 0.382 | 1 | – |
| PM | 0.474 | 0.645 | 0.560 | 0.385 | 1 |

Independent indicators—it is widely recognized that the relationship between corporate operating performance and corporate financial crisis is significantly positive. Thus, the chosen indicators in

financial crisis assessment must be reliable in order to become the surrogate for joining up with the independent indicators. The independent indicators implemented herein can be divided into two main categories: (1) numerical financial ratios derived from financial statements and (2) textual indicators derived from CSR reports. To extract the inherent information from multi-dimensional CSR reports, we executed LDA to break down these reports into some manageable dimensions. Perplexity is a commonly used measure in information theory to determine the optimal number of topics as well as to represent how well the model describes a dataset. The lower the perplexity is, the more superior the probabilistic model is [35]. According to the result (see Figure 3), we preserved four topics: environment–business-related dimension (EBRD), product–business-related dimension (PBRD), community–business-related dimension (CBRD), and customer–supplier-related dimension (CSRD).

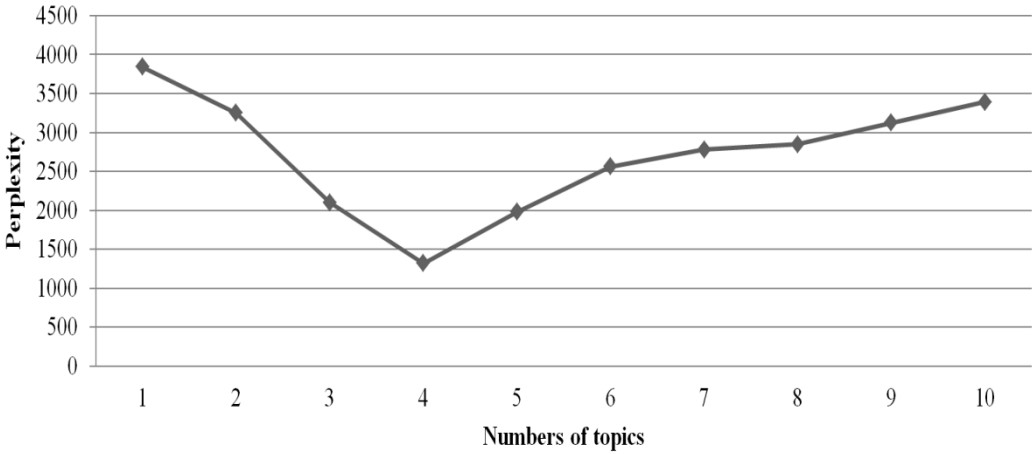

**Figure 3.** The perplexity result.

We sequentially matched each dimension's textual information with the pre-determined CSR-related corpus to convert a large amount of textual information into numerical information, as well as to eliminate computational cost. The CSR-related corpus was constructed by the joint utilization of the Chi-square approach ($x^2$) and entropy.

The Chi-square approach ($x^2$) was first used to extract the representative special terms from both categories (i.e., good CSR performance and bad CSR performance), which has demonstrated its usefulness in text categorization [59]. Entropy has been successfully implemented in computational linguistic analysis as a measure of the relative weights among all special terms [50], and thus it was conducted to determine the strength of each term. After finishing all the processes, we established the CSR-related corpus (i.e., a weighted list of words) (see Table 2). This corpus consists of two parts: one makes up the optimistic terms with a specific weight within the good CSR performance category, and the other encompasses the pessimistic terms with a specific weight within the bad CSR performance category. Table 3 presents the independent indicators used in this study.

**Table 2.** The partial of CSR-related corpus.

| With Good CSR Performance | | With Bad CSR Performance | |
|---|---|---|---|
| Term | weight | Term | Weight |
| Healthy | 0.090 | Loss | 0.083 |
| Continue | 0.071 | Affection | 0.067 |
| Responsibility | 0.064 | Decrease | 0.058 |
| Environmental protection | 0.055 | Casualties | 0.057 |
| Development | 0.051 | Deficiency | 0.052 |
| Assure | 0.042 | Malpractice | 0.048 |
| Promotion | 0.040 | Shock | 0.043 |
| Goal | 0.040 | Modification | 0.037 |
| Operation | 0.039 | Reduce | 0.030 |
| Performance | 0.037 | Lay off | 0.027 |
| Sustainability | 0.032 | Severance | 0.020 |
| Active | 0.028 | Indemnification | 0.018 |
| Efficiency | 0.021 | Illegal | 0.015 |
| Energy saving | 0.019 | Risk | 0.011 |
| Creativity | 0.013 | Bias | 0.006 |

**Table 3.** The independent indicators.

| Financial Related Ratio (Quantitative Ratios) | |
|---|---|
| X1: AR/S | Account receivable to sales |
| X2: S/TA | Sales to total assets |
| X3: TD/TA | Total debts to total assets |
| X4: I/TA | Inventory to total assets |
| X5: NP/TA | Net profit to total assets |
| X6: WC/TA | Working capital to total assets |
| X7: GP/TA | Gross profit to total assets |
| X8: EBIT/E | Earnings before interest and tax to equity |
| X9: AR/TA | Account receivable to total assets |
| X10: NFA/TA | Net fixed assets to total assets |
| X11: RE/TA | Returned earnings to total assets |
| **CSR Related Ratio (Qualitative Ratios)** | |
| X12: EBRD | Environment–business |
| X13: PBRD | Product–business |
| X14: CBRD | Community–business |
| X15: CSRD | Customer–supplier |

Figure 4 illustrates the difference of each CSR-related dimension between a corporation with superior operating status and a corporation with inferior operating status. The result reveals that a corporation with superior operating performance played a satisfactory job in CSR performance among all assessing criteria. This finding corresponds to the previous work done by McGuire et al. [60] who stated that a corporation with better CSR performance may face relatively lower capital constraints, fewer labor problems, and consumers who may be favorably disposed to its products as well as enhance its economic benefits. Furthermore, a corporation with better CSR performance may also have a lower financial risk owing to sustaining much more stable relations with the public sectors and financial communities.

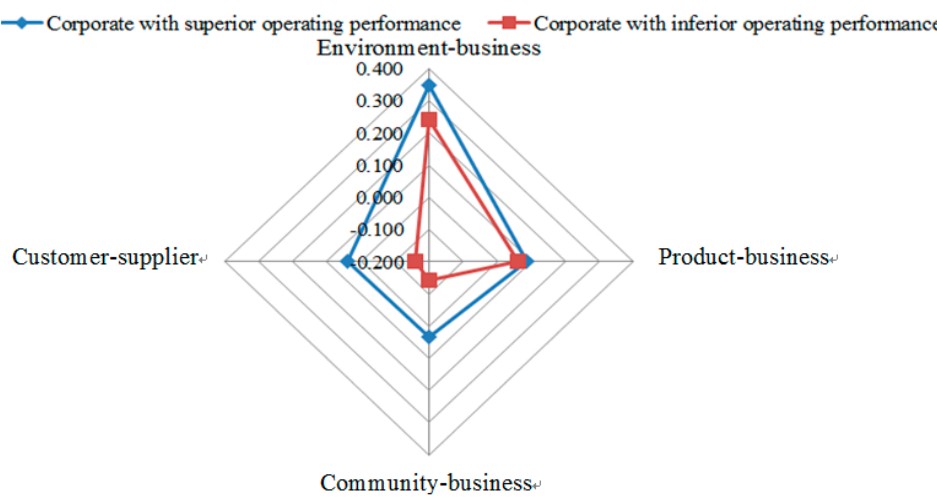

**Figure 4.** The CSR-related results.

### 3.3. The Practical Results

Before forecasting model construction, the feature subset selection technique was executed by FRST, which is an inevitable data pre-processing procedure in data mining and pattern recognition domains. Figure 5 depicts the selected features.

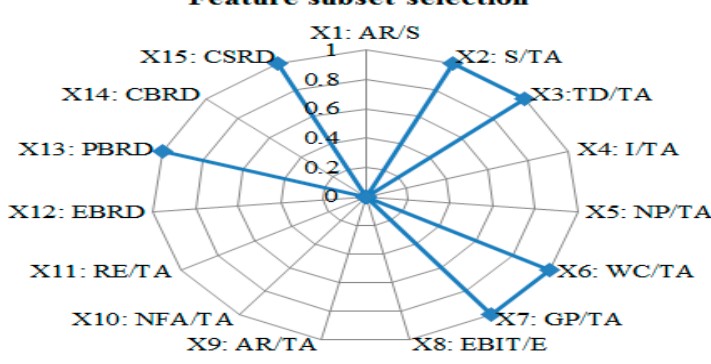

**Figure 5.** The selected feature subsets.

One of the interesting findings is that the selected features include CSR-related indicators. To examine the effectiveness of CSR-related indicators, this study set up two scenarios: (1) the model with CSR-related indicators and (2) the model without CSR-related indicators. There are many performance measures proposed for classification, with the commonly implemented performance measures being accuracy and error rate. However, merely performing one assessment criterion to determine the quality of the forecasting model is not reliable and trustworthy. Thus, this study adopted two other performance measures: sensitivity and specificity. The confusion matrix (see Table 4) and performance measures are illustrated as follows.

Overall accuracy: OA = (TN + TP)/(TP + FP + FN + TN)
Sensitivity: Sen = (TP)/(TP + FN)
Specificity: Spe = (TN)/(TN + FP)

**Table 4.** The confusion matrix.

| ↓Forecasted/Actual→ | Superior | Inferior |
| --- | --- | --- |
| Superior | True positive (TP) | False positive (FP) |
| Inferior | False negative (FN) | True negative (TN) |

Table 5 lists the forecasting results. To ensure the outcomes did not happen by chance, this study evaluated the significance of these results through a statistical examination. The Wilcoxon signed-ranks test, a non-parametric alternative to the paired t-test, was undertaken. It assumes commensurability of difference, but merely qualitatively: greater differences still count more, which are probably desired, but their absolute magnitudes are ignored [61]. From the aspect of statistics, this test is safer than the paired t-test since it does not assume normal distributions. Furthermore, the outliers have a lesser influence on the Wilcoxon signed-ranks test than on the paired t-test [61]. Table 6 depicts the result. We can see that the model going through the feature selection procedure posed higher forecasting performance (see Model A vs. Model C; Model B vs. Model D). It was because the feature selection posed numerous benefits, such as decreasing a model's forecasting bias, alleviating the storage requirement, and facilitating data visualization ability. The result also revealed that the forecasting model with CSR-related information not only had higher forecasting performance, but also played a satisfactory role in the other two assessing measures (see Model C vs. Model D). This finding is in accordance with Huang et al. [62], who stated that textual information can bring salient and incremental inherent information beyond a numerical aggregation from financial statements. Even a small improvement in forecasting quality can translate into large future savings for market participants. Due to the essence of textual information, the forecasting model construction should take this kind of information into consideration in order to reach a win–win station [17,63]. According to all of the experimental designs, the proposed model is a promising alternative for corporate operating performance forecasting. Decision makers can take it as a decision support system to allocate their valuable resources to reliable targets and to assist in forming their final decisions.

**Table 5.** The forecasting results (Avg. %).

| Criteria | Model A | Model B | Model C | Model D |
|---|---|---|---|---|
| Overall Accuracy | 68.17 | 75.29 | 79.03 | 84.01 |
| Sensitivity | 64.88 | 71.54 | 78.86 | 85.37 |
| Specificity | 70.60 | 78.07 | 79.16 | 83.01 |

Model A: With merely financial ratios (without feature selection) (X1~X11). Model B: With financial ratios and CSR-related ratios (without feature selection) (X1~X15). Model C: With merely financial ratios (with feature selection) (X2, X3, X6, X7). Model D: With financial ratios and CSR-related ratios (with feature selection) (X2, X3, X6, X7, X13, X15).

**Table 6.** The statistical results.

| Measure | Benchmark | Comparison | Hypothesis: $H_0:med_{diff} = 0$ $H_1:med_{diff} \neq 0$ |
|---|---|---|---|
| Overall Accuracy | Model D (84.01) | Model A (68.17) | 0.043 |
| | | Model B (75.29) | 0.043 |
| | | Model C (79.03) | 0.043 |
| Sensitivity | Model D (85.37) | Model A (64.88) | 0.043 |
| | | Model B (71.54) | 0.043 |
| | | Model C (78.86) | 0.043 |
| Specificity | Model D (83.01) | Model A (70.60) | 0.043 |
| | | Model B (78.07) | 0.068 |
| | | Model C (79.16) | 0.042 |

*3.4. Robustness Examination*

A large number of previous related works choosing a model with the best forecasting performance actually merely rely on one pre-decided database. It is a widely accepted idea that different assessment measures will result in different research outcomes. To strengthen the final decision we made, we tested the proposed model with another database. The variable of return on equities (ROE) was taken

as a performance measure in the new database. The outcome is in Table 7. We can easily see that the proposed model still outperforms the other three mechanisms under whole assessment criteria.

**Table 7.** The outcomes.

| Measure | Benchmark | Comparison | Hypothesis: $H_0{:}med_{diff} = 0$ $H_1{:}med_{diff} \neq 0$ |
|---|---|---|---|
| Overall Accuracy | Model D (86.33) | Model A (70.31) | 0.043 |
| | | Model B (78.35) | 0.043 |
| | | Model C (81.42) | 0.043 |
| Sensitivity | Model D (87.15) | Model A (72.02) | 0.042 |
| | | Model B (79.52) | 0.043 |
| | | Model C (82.31) | 0.043 |
| Specificity | Model D (85.73) | Model A (69.82) | 0.043 |
| | | Model B (77.43) | 0.043 |
| | | Model C (81.52) | 0.042 |

The introduced model could also be extended to a classifier ensemble. The basic idea of a classifier ensemble is to take advantage of various information outcomes by multiple dissimilar case classifiers instead of using the outcome of a traditional single classifier, so that forecasting performance can be facilitated and a biased decision can be prevented [64]. The consensus of domain experts attests to the effectiveness of a classifier ensemble resting on the preciseness and diversity of the base classifier [65,66].

To increase the range of the proposed model, this study executed two approaches: (1) kernel type diversity and (2) inherent parameter diversity. To make a fair comparison, this study took the proposed model with an ensemble structure as a benchmark and compared it with other classifier ensembles for operating performance forecasting, such as a neural network ensemble (NNE) [67] and support vector machine ensemble (SVME) [68]. Table 8 lists the test result, which proves that the proposed model performs a satisfactory job in forecasting the task with higher quality. No matter which structure the proposed model belongs to (individual or ensemble), it is indeed a promising alternative for corporate operating performance forecasting.

**Table 8.** The comparison results.

| Measure | Benchmark | Comparison | Hypothesis: $H_0{:}med_{diff} = 0$ $H_1{:}med_{diff} \neq 0$ |
|---|---|---|---|
| Overall Accuracy | LDA_ELM_SA Ensemble (88.73) | NNE (78.21) | 0.042 |
| | | SVME (81.52) | 0.042 |
| Sensitivity | LDA_ELM_SA Ensemble (89.62) | NNE (79.42) | 0.042 |
| | | SVME (82.31) | 0.043 |
| Specificity | LDA_ELM_SA Ensemble (87.31) | NNE (77.62) | 0.042 |
| | | SVME (81.63) | 0.043 |

To further examine the explanation ability of CSR-related ratios, we put the selected ratios into the regression model. We can see that the model with CSR-related ratios had a higher explanation ability than the model without CSR-related ratios. The results (see Table 9) provide evidence for the usefulness of textual information.

**Table 9.** The results.

| Model | r-Square | Adjusted r-Square |
|---|---|---|
| Regression model with CSR-related ratios | 0.685 | 0.672 |
| Regression model without CSR-related ratios | 0.542 | 0.531 |

*3.5. Discussion*

For more than three decades of CSR discussions and debates in previous research works, no single and conclusive definition of this concept exists [69]. One definition comes from the European Commission, in which CSR as a "concept whereby firms integrate social and environmental concerns in their business operations and in their interaction with their stakeholders on a voluntary basis" appears to be the most suitable description of it. The relationship between CSR and corporate financial performance turns out to be the most widely studied topic in both business and society literature [70–72]. Many empirical studies have been conducted over the last three decades, but still do not reach consistent results [73]. Follow-up review studies and empirical works [74,75] conclude on balance that there is a positive, yet mild, relationship between CSR and a firm's financial performance. Despite this expanding stream of literature, academics and practitioners are still uncertain when it comes to assessing which specific dimension of CSR is the most effective toward a firm's financial performance [72,73].

To fill the gap in both the CSR and financial performance literature, this study decomposed CSR into numerous dimensions by LDA and further examined each CSR dimension's influence on a firm's financial performance. LDA is a data-driven technique without human intervention (that is, it tries to let words speak). Thus, the results derived from LDA are much more objective. As such, firm managers can realize which dimension helps facilitate the firm's financial performance and re-allocate resources to suitable places.

The research targets in this paper were resource-abundant publicly-listed firms. The managers of these firms are likely to undertake, for example, an employee training program or a boarder community program, because doing so would generate some benefits for the firm's future prospects [73]. On the other hand, managers of resource-scarce, non-publicly-listed firms (small- and medium-sized enterprises, SMEs) have greater chances to postpone such investments and prioritize more pressing requirements [76]. This is consistent with previous research work done by Panwar et al. [73], who indicated that the firm's level of CSR highly relates to its financial resources. To gain a more holistic perspective of CSR's impact on a firm's financial performance, the study should have taken publicly-listed and non-publicly-listed firms into consideration. However, due to a lack of availability of non-publicly-listed firms' data, this study just considered publicly-listed firms' data to reach our research findings.

Clearly distinguishing between CSR dimensions makes it easier and more tangible for decision makers to assess CSR engagement's benefits by focusing on their firm's stakeholders: customers, employees, suppliers, shareholders, environment, and the local community [77]. Phole and Hittner [78] indicated that a qualitative research method seems to be a proper approach to depict the relationships between CSR dimensions. Thus, much research relied on solid qualitative methods to determine the components of CSR [77]. The customer domain encompasses issues such as fair prices, clear labeling, safe and high-quality products, etc. The employee domain addresses issues like appropriate remuneration, reliable promotion systems, and good working conditions. The supplier domain emphasizes the topic of the fairness of issues like supplier determination, internal auditing, product conditions, and shipping contracts. The shareholder domain addresses topics that generate profit and maximize personal wealth. The environment domain consists of topics such as a reduction in polluted products and wastes, eliminating toxic chemicals, and saving energy. The local community covers topics such as creating jobs for people living in the community, making contributions to a region's development, and sourcing locally. All addressed domains are included in CSR. However, this study just considered four dimensions of CSR (environment–business-related dimension: EBRD, product–business-related dimension: PBRD, community–business-related dimension: CBRD, and customer–supplier-related dimension: CSRD) by performing a perplexity score and further

examined each dimension on a firm's financial performance. To reach an overarching and robust conclusion, future studies can take more dimensions into consideration and examine each dimension's impact on a firm's financial performance.

## 4. Conclusions and Implications

Prior related works have yielded non-conclusive results regarding the relationships between corporate operating performance and corporate social responsibility. Those conflicting results may be attributed to two differences in measuring financial performance and CSR status [32,79]. Accounting-based ratios, such as ROA and ROE, are the most commonly implemented to determine corporate operating performance, but they are categorized into the domain of one input variable and one output variable. However, employing only one input variable and one output variable to determine such performance is not reliable in this highly fluctuating economic atmosphere. To overcome the obstacle, this study executed DEA as it can handle multiple input variables and multiple output variables simultaneously.

A firm's CSR policy is multi-dimensional and includes numerous aspects, such as environmental, business, and social factors [79]. Performing merely one general indicator to depict a corporation's CSR status could cause some uncertainties. Thus, to better understand the relationship between CSR and corporate financial performance, this study opened up the black-box of CSR (i.e., breaking it into numerous dimensions by LDA) and examined the impact of each CSR dimension on corporate financial performance. Doing so allows top executives to allocate limited resources to specific CSR dimensions that are highly related to corporate financial performance.

In contrast with well-studied works such as financial crisis prediction, research on corporate operating performance is quite rare. It is widely deemed that a corporation with worse operating performance could be at the stage before a financial crisis bursts out. If top executives can receive this red flag signal (that is, crisis warning signal) earlier, then they can initiate a modification of their firm's capital structure and adjust investment strategies so as to enhance its risk absorbing ability. To our knowledge, there is no study in the literature that has introduced a model for corporate operating performance forecasting that simultaneously considers multiple dimensions of CSR broken down by LDA and uses DEA as the performance measure. To fill this gap in the CSR and corporate finance literature, this study proposed an emerging corporate operating performance forecasting model that consists of multiple dimensions of CSR, feature subset selection by FRST, and model construction by ELM_SA. The proposed model, examined through real cases, is a promising alternative for corporate operating performance.

This study had some limitations that could be addressed in future studies. First, we worked on the target sample of Taiwan's electronic industry, which suggests that the ability to generalize the results is limited. Second, future research can enhance the model's forecasting quality by integrating much more sophisticated structures, such as ensemble learning and stacking [80,81]. Finally, recent related works suggest that the debate concerning corporate operating performance and CSR status should be taken further by including additional intermediate variables or the variables derived from theoretical constructs that can enhance our realization of the procedures through which CSR influences corporate operating performance [72,73,76,82].

**Author Contributions:** Conceptualization, Y.-L.H. and M.L.; methodology, M.L.; software, M.L.; validation, Y.-L.H.; formal analysis, Y.-L.H.; investigation, Y.-L.H. and M.L.; resources, Y.-L.H.; data curation, Y.-L.H.; writing—original draft preparation, Y.-L.H. and M.L.; writing—review and editing, Y.-L.H. and M.L.; visualization, Y.-L.H. and M.L.; supervision, M.L. All authors have read and agreed to the published version of the manuscript.

**Funding:** This research received no external funding.

**Conflicts of Interest:** The authors declare no conflict of interest.

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
