# Peer review of "Corporate Social Responsibility and Corporate Performance: A Hybrid Text Mining Algorithm"

_sustainability, doi:10.3390/su12083075_

Round 1

Reviewer 1 Report

This is a well conceptualized and executed study. The main concern is to the degree it adds new knowledge that will advance CSR as a field of study. 

One comment I would add is that 99% of the firms in the world are small firms and their financial data is not publicly available. I would urge you to include in the discussion section how your results may or may not help advance our understanding about small firms CSR. You can use the following paper to couch your discussion:

"Does the business case matter? The effect of a perceived business case on small firms' social engagement." 2017. Journal of Business Ethics 144: 597-608. 

Author Response

Dear Reviewer:

I have carefully revised this manuscript followed your valuable suggestions.

Reviewer 2 Report

The paper presents an interesting and complex methodology for performance prediction. Nevertheless, it neglects important literature about CSR measurement. There is a growing literature that analyzes different dimensions of CSR and tests it according to the perception of different stakeholders, like those studies I cite below. Moreover, surrounding theories about what are the dimensions must be not neglected in order to define it. Theory must be the basis for analyzing and defining the measure.

Examples of studies that develop CSR measures:

Martínez, P., Pérez, A. and Rodríguez del Bosque, I. (2013) ‘Measuring corporate social responsibility in tourism: development and validation of an efficient measurement scale in the hospitality industry’, Journal of Travel & Tourism Marketing, Vol. 30, No. 4, pp.365–385.

Öberseder, M., Schlegelmilch, B.B., Murphy, P.E. and Gruber, V. (2014) ‘Consumers’ perceptions of corporate social responsibility: scale development and validation’, Journal of Business Ethics, Vol. 124, No. 1, pp.101–115.

Pérez, A., Martínez, P. and Rodríguez del Bosque, I. (2013) ‘The development of a stakeholderbased scale for measuring corporate social responsibility in the banking industry’, Service Business, Vol. 7, pp.459–481.

Pérez-Aranda, J.A., and Boronat-Navarro, M. (2020 in press) Which corporate social responsibility issues do consumers perceive as relevant to be evaluated in the hotel sector?, European Journal of International Management.

Author Response

Dear Reviewer:

I have carefully revised this manuscript followed your valuable suggestions.

Thank you very much!

Round 2

Reviewer 2 Report

The authors make an effort to cope with my concerns. With the explanation made in section 3.5, the paper is most understandable. Nevertheless, I still think that you may state in the introduction that your development is a data-driven measure, that your aim is not to develop the measurement of a theoretical construct (like in the studies I mentioned in my previous review). Therefore, your paper is different to those that are traying to derive a measure for a theoretical construct. Therefore, as you now stated in section 3.5, there are some dimensions that are not in your performance-measure. You could acknowledge the studies that develop the theoretical construct, and also, you would mention this difference in conclusions when you state study limitations.

Author Response

Dear Reviewer:

Thanks for your valuable and useful comments, we have re-structured the manuscript carefully.

Please see the attached files.

Thank you very much.
